

# The role of a low-level jet for stirring the stable atmospheric surface layer in the Arctic

Ulrike Egerer[1, now at 3], Holger Siebert[1], Olaf Hellmuth[1], and Lise Lotte Sørensen[2]

[1]Leibniz Institute for Tropospheric Research, Permoserstr. 15, 04318 Leipzig, Germany
[2]Aarhus University, Frederiksborgvej 399, 4000 Roskilde, Denmark
[3]National Renewable Energy Laboratory (NREL), 15013 Denver W Parkway, Golden, CO 80401, USA

**Correspondence:** Ulrike Egerer (ulrike.egerer@nrel.gov)

**Abstract.** In this study, we analyze the transition of a stable atmospheric boundary layer (ABL) with a low-level jet (LLJ) to a traditional stable ABL with a classic Ekman helix in the late-winter central Arctic. Vertical profiles in the ABL were measured with a hot-wire anemometer on a tethered balloon during a 15 h period in March 2018 in northeast Greenland. The tethered balloon allows high-resolution turbulence observations from the ground to the top of the ABL. The core of the LLJ was observed at about 150 m altitude, and its height and strength were associated with the temperature inversion. Increased turbulence was observed in the vicinity of the LLJ, but most of the turbulence does not reach down to the surface, thus decoupling the LLJ from the surface. Only when the LLJ collapses and the ABL again exhibits a more classical Ekman spiral, a coupling to the surface is re-established. Numerical simulations using an analytical model support these observations and allow conclusions to be drawn about the possible role of an LLJ in the advection of a passive tracer such as aerosol particles or moisture.

## 1 Introduction

Low-level jets (LLJs) are vertically more or less bounded wind fields with local maximum wind velocities exceeding the geostrophic wind. They can be observed in the stably stratified atmospheric boundary layer (ABL) in heights below 1 km or so and are related to inertia oscillations and a decoupling of the flow dynamics from the surface-layer friction (Blackadar, 1957; Smedman et al., 1993). There is a lively debate about the origin of LLJs (Vihma et al., 2011; Jakobson et al., 2013; Tuononen et al., 2015; Guest et al., 2018; Chechin and Lüpkes, 2019) and details of the mechanism behind this are still not completely understood. Polar regions are preferable locations for the occurrence of LLJs (Tuononen et al., 2015; López-García et al., 2022) due to frequently observed (extremely) stably stratified and shallow ABLs, in particular during cloudless (wintertime) conditions with strongly negative thermal-infrared net irradiance.

The stably stratified boundary layer, as defined by an increase of potential temperature with height, usually exhibits comparably low turbulence. When neglecting cloud-related effects such as radiative cooling at cloud top, the only significant source of turbulent kinetic energy (TKE) is the surface roughness. However, due to the strong local wind shear below and above the wind maximum, an LLJ can introduce a significant amount of TKE (Smedman et al., 1993) and modulate the vertical distribution of TKE (Jakobson et al., 2013; Banta et al., 2006). These wind shear zones are usually considered a source of TKE production



as long as the damping effect of the inversion is not dominating over the TKE production. Boundary-layer flows with a source of turbulence at the top of the inversion generated by vertical wind shear have also been referred to as "upside down" (Mahrt, 1999). The relative importance of turbulence-generating wind shear and damping inversion is quantified by the dimensionless Richards number (Ri) defined later on. In addition to TKE production, LLJs may play an important role in the advection of momentum, turbulence, as well as aerosol particles and precursor gases (Stensrud, 1996; Algarra et al., 2019).

Here, we study the ABL evolution in terms of an LLJ, as observed by a set of subsequent vertical wind and temperature profiles measured with a tethered balloon in the framework of the Polar Airborne Measurements and Arctic Regional Climate Model SImulation Project (PAMARCMiP) at Station Nord in northeast Greenland. The central question of this work is how the LLJ might affect the vertical distribution of turbulence and whether increased turbulence is observed at the surface when the LLJ occurs or collapses. This could indicate that properties advected with the LLJ, such as increased aerosol concentration

or precursor gases, could also be mixed down to the surface after being advected over a certain distance inside the LLJ. Accompanying numerical simulations based on an analytical model support these studies and allow reasoning on the role of an LLJ in advection and subsequent vertical mixing during the collapse of the LLJ.

## 2    Observations

### 2.1    PAMARCMiP campaign

The PAMARCMiP field campaign was conducted at the Villum Research Station (VRS) on the military outpost 'Station Nord' in the northeast corner of Greenland on the small peninsula of Princess Ingeborg ($81°36'$N, $16°40'$W). Balloon observations were made from 10 March to 7 April 2018 about 2 km south of VRS at a small observation hut ('Flygers Hut') to minimize the station's influence. The site is located on the coastline between the Greenland Ice Sheet to the south and the Arctic Ocean, which is covered with sea ice for most of the year at this location. A glacier is located about 20 km to the south.

The LLJ and ABL structure observations are mainly based on the tethered balloon system BELUGA (Egerer et al., 2019). Continuously running meteorological measurements at 9 m altitude at the VRS and three-dimensional wind data from an ultrasonic anemometer (USA-1, manufactured by METEK GmbH, Germany) mounted on a tower at 65 m height, support the analysis of the profile measurements.

### 2.2    The BELUGA setup

BELUGA is a modular system consisting of tethered balloons of various sizes and a variety of sensor packages. During PAMARCMiP, a 9 $m^3$ balloon with a maximum payload of 3 kg was used with multiple sensor packages. In this work, data from two sensor packages are used: (i) A turbulence probe based essentially on a single-component hot-wire anemometer measuring wind speed with a sampling frequency of 500 Hz. A Pitot-static tube served as a reference for the hot-wire sensors. Fast temperature measurements were made with a cold-wire sensor. In addition to a calibrated thermometer as a reference

for the cold-wire sensor, the data logger provided additional pressure measurements for barometric altitude. (ii) A standard



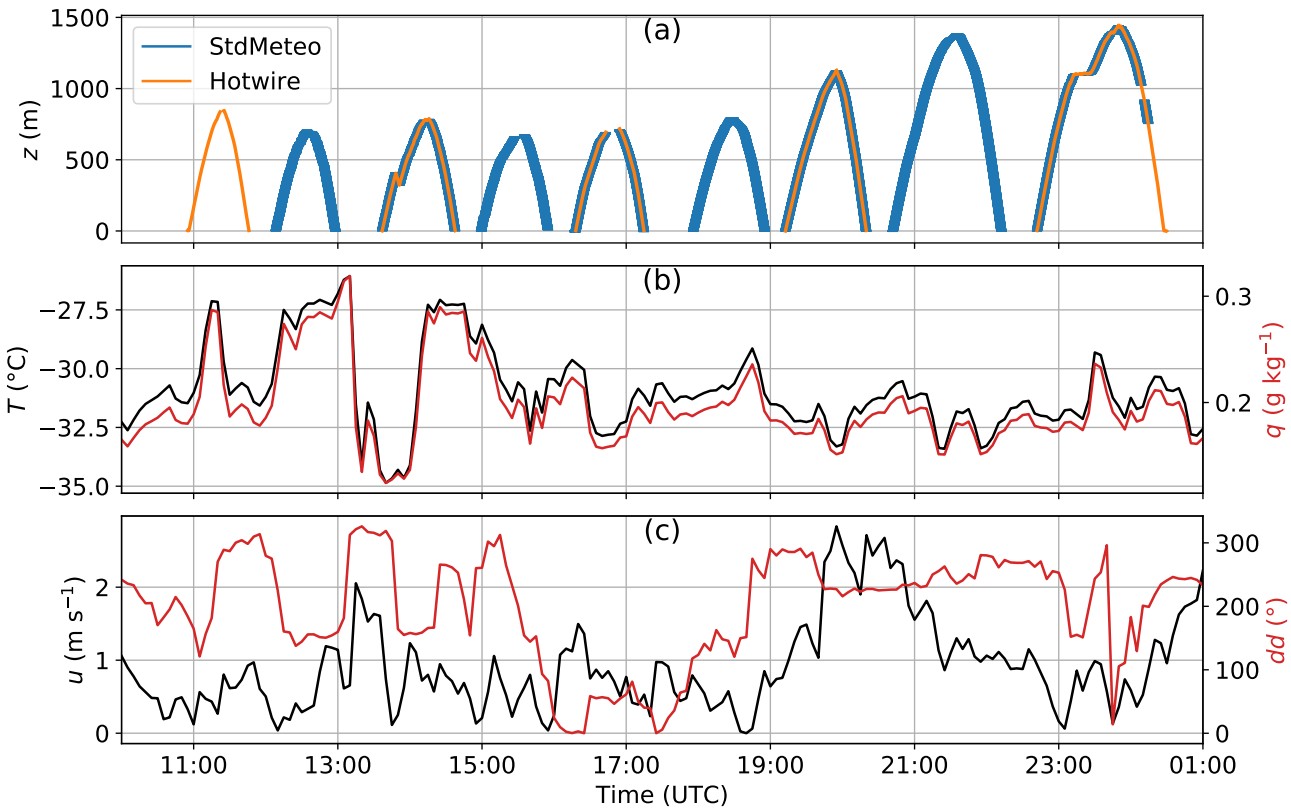

**Figure 1.** Time series for 29/30 March 2018 of (a) BELUGA flight altitude and (b) VRS near-surface measurements of temperature $T$ and specific humidity $q$ as well as (c) wind velocity $u$ and wind direction $dd$.

meteorological probe based on a Graw DFM-09 radiosonde with an additional Pitot probe provided wind speed and direction using a compass. This package measured wind, temperature, and relative humidity at a frequency of 1 Hz.

## 2.3 Measurements and synoptical situation

This study focuses on nine subsequent flights between 10:00 UTC on 29 March and 1:00 UTC on 30 March 2018, during the

60 transition between polar night and day. The period was mainly characterized by the persistent occurrence of an LLJ, followed by a transition to a more classic stable ABL. Synoptic conditions during this period were influenced by high air pressure over central Greenland yielding calm weather conditions at ground level. All flights occurred under cloudless conditions with only thin clouds well above 1000 m. Therefore, possible icing was not an issue.

Figure 1 provides an overview of the nine BELUGA flight profiles up to altitudes of 600 to 1400 m in combination with

65 near-surface measurements. The near-surface conditions are quite variable in terms of temperature $T$ and specific humidity $q$, with $T$ ranging between -35 °C and -26 °C and $q$ ranging between 0.15 to 0.3 g kg$^{-1}$. Qualitatively, there is an obvious high





correlation between $T$ and $q$. Wind speeds $u$ were below $2\ \mathrm{m\,s^{-1}}$ and decreased towards the end of the observation period. The wind direction $dd$ is mainly from the west, but turns over south to north from 16:00 UTC and back to west after 19:00 UTC. Between 13:00 and 14:00 UTC, $T$ and $q$ drop abruptly by -8 K and $0.2\ \mathrm{g\,kg^{-1}}$, respectively, along with a change in wind
direction (southeast to north) and reach their previous values about one hour later.

## 3   Data analysis methods

In this study, we use the balloon-borne vertical profiles to relate the properties of an LLJ to the vertical structure of stability and turbulence. The literature reveals various definitions of an LLJ, mostly based on a local low-altitude wind velocity maximum greater than around $2\ \mathrm{m\,s^{-1}}$ (Tuononen et al., 2015; Andreas et al., 2000; Blackadar, 1957). We adopt this definition with the
following criteria for an LLJ: (i) the wind velocity maximum occurs below 250 m altitude and (ii) the difference between the maximum wind velocity in the LLJ core $u_{\mathrm{LLJ}}$ and the wind minimum $u_{\mathrm{min}}$ above *and* below (commonly the near-surface wind velocity) exceeds $2.5\ \mathrm{m\,s^{-1}}$. The LLJ strength is then defined as $\Delta u = u_{\mathrm{LLJ}} - u_{\mathrm{min}}$ with $u_{\mathrm{min}}$ being the higher value either above or below the jet core. The height of the LLJ core $z_{\mathrm{LLJ}}$ is typically located at the maximum height of the (strong) temperature inversion $z_i$.
The dimensionless gradient Richardson number $\mathrm{Ri}_g$ is the ratio of buoyancy to shear:

$$\mathrm{Ri}_g = \frac{g}{\overline{\theta}} \cdot \frac{\partial\theta/\partial z}{(\partial u/\partial z)^2}\ , \tag{1}$$

with the vertical gradient of the mean flow speed $(\partial u/\partial z)^2 \approx (\partial u_x/\partial z)^2 + (\partial u_y/\partial z)^2$, the vertical potential temperature gradient $\partial\theta/\partial z$, and the acceleration of gravity $g$. When stratification dominates over wind shear and a critical Richardson number $\mathrm{Ri}_c$ - estimated to be between 0.25 and 1 (Miles, 1961; Abarbanel et al., 1984) - is reached, turbulence decreases.
Thus, the contribution of buoyancy and shear to the turbulence profile throughout the ABL can be analyzed by the shear and buoyancy terms of the $\mathrm{Ri}_g$. For estimating $\mathrm{Ri}_g$, $u$ and $\theta$ profiles are smoothed with a 10 s rolling mean before calculating the local gradients. The final $\mathrm{Ri}_g$ profile is then smoothed again with a rolling mean with a 10 s window. As an approximation for the range between the surface and the core of the LLJ, the bulk Richardson number $\mathrm{Ri}_b$ (Mahrt, 1985) is calculated for the height $z$:

$$\mathrm{Ri}_b(z) = \frac{g}{\overline{\theta}} \cdot \frac{\Delta\theta(z) \cdot z}{\Delta u^2(z)}. \tag{2}$$

Here, $\Delta\theta(z)$ (or $\Delta u^2(z)$) is the difference between $\theta$ (or $u$) at $z$ and close to the surface.

The local turbulence is characterized by means of the local energy dissipation rate $\varepsilon$ which is derived from the second-order structure function of $u$ applying inertial sub-range scaling:

$$\overline{(u(t-t^*) - u(t))^2} = C \cdot \varepsilon^{2/3} \cdot (t^* \cdot \overline{u})^{2/3} \tag{3}$$

with $C = 2$ and a time lag $t^*$ for the longitudinal wind velocity $u$ (Siebert et al., 2006; Wyngaard, 2010). Here, the overline in Eq. (3) denotes time averaging; an averaging interval of 2 s was selected which yields robust estimates (Egerer et al., 2019). The minimum resolvable $\varepsilon$ is estimated to be about $10^{-6}\ \mathrm{m^2\,s^{-3}}$.



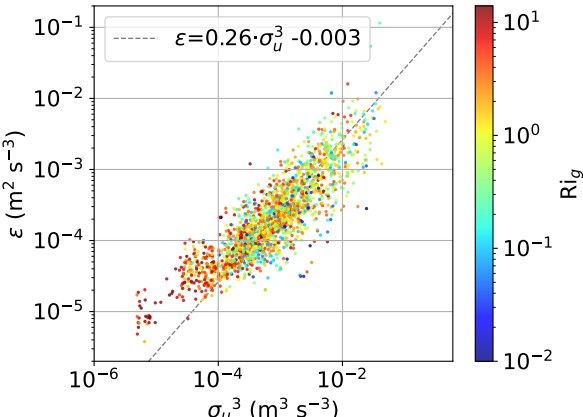

**Figure 2.** Relation of turbulence parameters for all BELUGA flights on 29 March 2018 with the hot-wire: Gradient Richardson Number $\mathrm{Ri}_g$, $\sigma_u^3$, and local dissipation rate $\varepsilon$. Each dot represents a 5 s time interval of averaged parameters. The dashed line shows a linear regression fitted to the data points.

To better classify both the integration and averaging times in the determination of parameters as discussed before, but also to better interpret the vertical profiles in the next sections, knowledge of the length scales involved is of certain interest. Typical
(integral) length scales of a turbulent flow can be defined according to

$$\mathcal{L} = \beta \cdot \sigma_u^3 / \varepsilon \qquad (4)$$

(Wyngaard, 2010) with a constant $\beta$ of $\mathcal{O}(1)$. Here, $\varepsilon$ is interpreted as a local parameter derived from 5s-long sub-records whereas $\sigma_u^3$ (with $\sigma_u$ being the standard deviation of $u$) is determined by the largest contributing scales and, therefore, is averaged over a larger window of 30 s. Figure 2 shows a scatter plot of $\varepsilon$ versus $\sigma_u^3$ based on all BELUGA data observed on 29
March 2018. For each measuring point, the corresponding $\mathrm{Ri}_b$ value is represented by a color code to better classify the values. The relation of observed $\varepsilon$ and $\sigma_u^3$ is almost linear as predicted by Eq. (4), but shows some scatter. The resulting local length scales $\mathcal{L}$ (assuming $\beta = 1$) are mainly in the range between 1 to 10 m (this applies to 85 % of the data points) with somewhat larger length scales in regions of comparably high turbulence. A few data points with $\mathrm{Ri}_b > 1$ (reddish color) indicate small length scales below around 1 m. The ratio of the assumed mean wind velocity of $\overline{u} \approx 5 \,\mathrm{m\,s^{-1}}$ to the averaging time of 30 s for
estimating $\sigma_u$ yields an averaging length of 150 m which is about 10 to 100 times larger than the $\mathcal{L}$ estimates. Therefore, we can safely conclude that the averaging period covers enough eddies and the results are statistically robust.

For the 65 m mast wind data, $\varepsilon$ is estimated analog to the vertical profiles, but averaged over 5 min segments and using the vertical wind velocity component. The vertical wind velocity of the 65 m mast sonic anemometer for the flux calculation is tilt-corrected using the double-rotation algorithm described by Wilczak et al. (2001). For the determination of the turbulent
fluxes of virtual sensible heat, $H = \rho \cdot c_p \cdot \overline{\theta_v' w'}$, and momentum, $\tau_u = \rho \cdot \overline{u' w'}$, the fluctuations are calculated by applying linear detrending and averaging over 30 min periods at 5 min time steps.



**Figure 3.** Time-height contour plots and time series observed at 9 m (green) and 65 m (blue) altitude for (a) wind velocity $u$, (b) wind direction $dd$, and (c) temperature $T$/ potential temperature $\theta$. The data are from three different sources: (i) The ascends and descends are from BELUGA, (ii) the data shown at 9 m altitude are from VRS, and (iii) the data at 65 m altitude are from the mast. The dashed vertical lines represent the time of the example flights in Fig. 4.



## 4 Observed vertical structure of the ABL and LLJ

### 4.1 Evolution of the LLJ and ABL structure

We first analyze the evolution of the mean ABL structure and the LLJ throughout the observation period. Figure 3 provides an
overview of the period based on BELUGA profiles (time-height contour plots), VRS observations in terms of meteorological
measurements at 9 m and the sonic data at 65 m height. The observation period can be divided into three sub-periods: (I) an
LLJ period, (II) a transition period, and (III) a standard stable ABL.

In the first period (10:00 to 16:00 UTC), a clear LLJ emerges with a wind velocity maximum of about $10\,\mathrm{m\,s^{-1}}$ in a height of
around 100 to 150 m, while the wind velocity in the near-surface layer remains below $2\,\mathrm{m\,s^{-1}}$. The wind direction in the lower
400 m is west to northwest with the highest variability at surface level. A very stable surface layer develops with near-surface
temperatures around -30 °C, which increase by 10 K to 100 m height. The sharp surface temperature drop at 13:15 is not
obvious at 65 m altitude and qualitatively correlates with a wind rotation to the north.

Between about 16:30 and 21:00 UTC, the wind velocity becomes more constant with height but generally decreases through-
out the profile from about $5\,\mathrm{m\,s^{-1}}$ to less than $2\,\mathrm{m\,s^{-1}}$ with a highly variable wind direction. The LLJ disappears almost
completely, and the strong surface temperature gradient now extends only to the lowermost tens of meters. At 17:00, however,
another smaller LLJ occurs with a maximum at a lower altitude compared to the previous LLJs of the first period. The temper-
ature throughout the entire profile decreases by about 5 to 10 K. This period is labeled as a transition between the LLJ period
and a standard stable ABL structure.

After 21:00 UTC, the wind velocity increases with a local maximum at around 100 m but is much less pronounced compared
to the LLJ structure observed in the first period. The wind direction is almost constant throughout the profile. The temperature
profile is similar to the transition phase with a temperature inversion situated at a lower altitude.

For a more detailed analysis of the stratification during the three periods, four selected individual profiles of $u$, $\theta$, and $\varepsilon$
are plotted in Fig. 4. A well-developed LLJ is observed at 14:31 UTC. The wind velocity maximum of $9\,\mathrm{m\,s^{-1}}$ in 100 m
coincides with the top of the strong surface-based temperature inversion. Above this inversion, the ABL is almost adiabatically
stratified up to 150 m - the region with decreasing wind velocity. This region between 100 m and 150 m shows the highest
local variability of wind velocity, although the wind shear is lower compared to the height range between 40 m and 100 m.
This observation is visible in the local energy dissipation, which is highest - apart from the lowermost surface layer - in the
upper part of the LLJ. Here, with almost neutral stratification, the wind shear term in Eq. (1) dominates the buoyancy term by
one order of magnitude. Below the LLJ core in 40 to100 m, the turbulence generation due to wind shear is reduced (compared
to above the LLJ) by the influence of the temperature inversion. A local minimum of $\varepsilon$ is located between the surface and the
base of the LLJ, where the damping effect of the strong temperature inversion coincides with a height-constant wind velocity.
Above 160 m, $\theta$ again slightly increases with height and $\varepsilon$ drops by two orders of magnitude.

The profile observed at 19:12 UTC in the transition phase shows the lowest wind velocity ($\approx 1\,\mathrm{m\,s^{-1}}$) between 50 m and
160 m. The maximum wind velocity is above $3\,\mathrm{m\,s^{-1}}$ in the near-surface layer, coinciding with a temperature inversion up
to 40 m. However, just above this inversion, there is a shallow, 20 m-thick layer where the wind shear is strong enough to





develop some turbulence indicated by a local maximum of $\varepsilon$ up to $10^{-3}$ $\mathrm{m^2\,s^{-3}}$. Above and below this maximum, stable stratification in combination with weak wind shear results in reduced turbulence. In the transition phase, one single profile includes an LLJ again, with a slightly different structure than the first period observations (Fig. 4, upper right). Here, the wind velocity increases with height directly above the surface with $u_{\mathrm{LLJ}} = 7\,\mathrm{m\,s^{-1}}$ at 60 m height. The lowermost 20 m are neutrally

stratified followed by a temperature inversion at $z_i = 100\,\mathrm{m}$, which coincides with the wind minimum above the LLJ. Here, the $\varepsilon$ profile is different from the LLJ cases, with maximum turbulence at the surface, gradually decreasing towards the upper bound of the LLJ, allowing turbulent mixing between the surface and the LLJ core. The upper part of the LLJ is still located inside the temperature inversion, leading to less turbulence in that region than for the LLJ phase. Above the LLJ, the turbulence intensity remains low and slightly increases at higher altitudes, with the wind velocity increasing again.

In the period with a standard stable ABL structure, the surface layer up to 50 m shows the largest increase of $u$ and $\theta$, thus resulting in generally low values of $\varepsilon$ compared to the previous periods. Above 50 m, the ABL is slightly stably stratified with an almost height-constant wind velocity, weakest turbulence at the top of the surface inversion, and some variable turbulence at higher altitudes. Throughout the profile, the values for buoyancy and shear increase or decrease to a similar degree, resulting in turbulence predominantly induced by surface roughness.

As a next step, we examine the temporal evolution of LLJ and ABL parameters. Figure 5 shows the time series of LLJ and temperature inversion strength and height, as well as $\mathrm{Ri}_b$ between surface and $z_i$ derived from the profiles, combined with continuously measured turbulence parameters at 65 m. Except for a few cases in the transition phase, the LLJ strength correlates with the temperature inversion strength (Pearson correlation coefficient $R = 0.52$). The LLJ height correlates even more clearly with the inversion height ($R = 0.64$). $\mathrm{Ri}_b$ is increased in the transition period and around Ri=1 (weak turbulence)

for the other two periods. Energy dissipation $\varepsilon$ in the LLJ period is one order of magnitude higher than in phases II and III and shows much more variability. The turbulent fluxes of heat $H$ and momentum $\tau_u$ show increased values and variability and individual events with high flux magnitude in the LLJ period, compared to low fluxes in the transition and standard ABL phase. The LLJ, as observed at 17:04 UTC coincides with a short period of weak upward-oriented momentum fluxes and downward heat fluxes.

## 4.2 Normalized vertical profiles

Each of the three periods introduced in Sect. 4.1 features a distinct vertical structure of thermodynamic and turbulence parameters. For each of the three periods, Fig. 6 shows normalized vertical profiles. Whereas the height is normalized with $z_i$, $u$ is normalized with $u_{\mathrm{max}}$ (below 300 m), and $\theta$ and $q$ are normalized with their values at $z_i$. The box plots are assigned to height intervals and include median values for each profile within the respective period. The wind velocity profile shows a charac-

teristic shape for the LLJ period and the standard stable ABL period, with low variability within the profiles. The transition phase exhibits much higher variability, and wind velocity increases with height. The $\theta$ and $q$ profiles are similar for all phases, with the lowest variability below $z_i$ for the LLJ period. In each period, there is a turbulence maximum in $\varepsilon$ close to the surface, which indicates surface-driven turbulence for all cases. In relation to the wind profile, the vertical turbulence profile is different for each period and will be discussed in more detail.





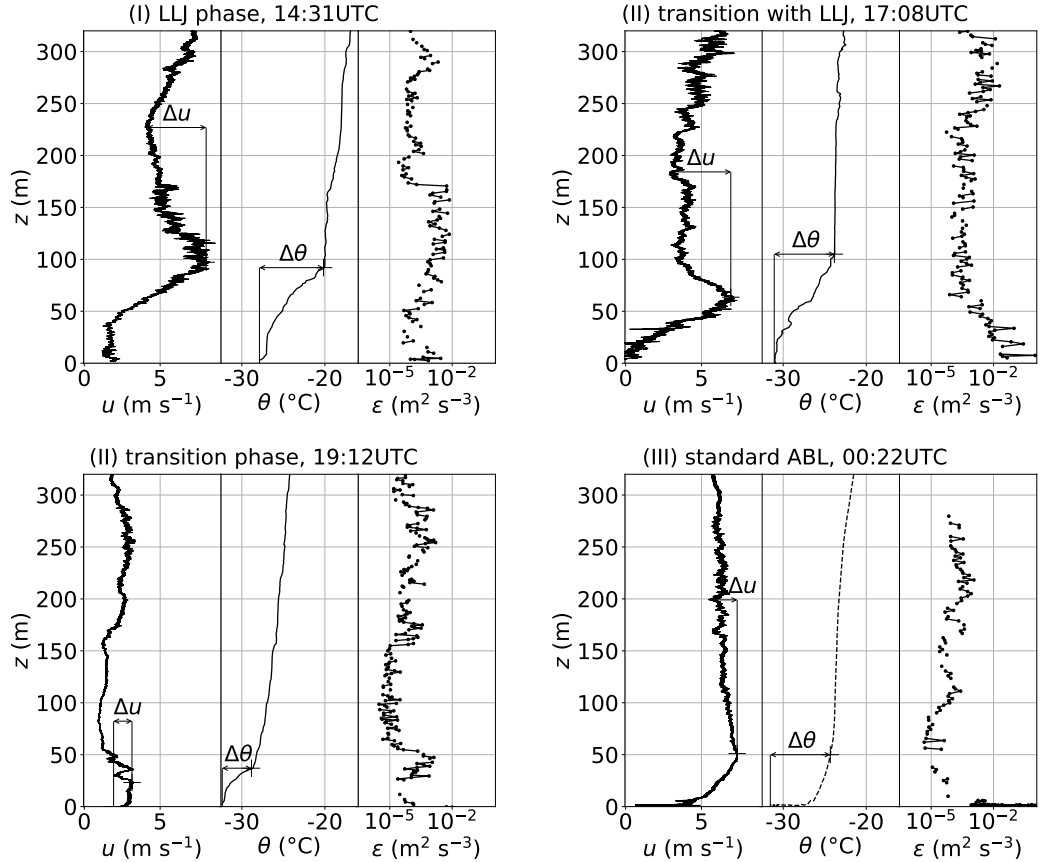

**Figure 4.** Example vertical profiles for each phase: wind velocity $u$ with the definition of LLJ strength $\Delta u$ and LLJ height $z_{\mathrm{LLJ}}$, potential temperature $\theta$ with the temperature inversion strength $\Delta \theta$ and inversion height $z_i$ and dissipation rate $\varepsilon$. For the transition phase, one profile with and one profile without LLJ is shown.

In the LLJ period, the average $\varepsilon$ structure has a characteristic shape with three local maxima: at the surface and just above and below $z_i$ with a local minimum around $z_i$ itself. Another local minimum of $\varepsilon$ at $z/z_i \approx 0.3$ suggests decoupling of the LLJ from the surface. In the transition period, the average $u$ profile is almost height-constant but shows high internal variability. The $\varepsilon$ profile is highly variable with height without any characteristic structure. The only common features are minimum values around $z_i$ and maximum values close to the surface.

The last period is characterized by a gradual increase of the normalized velocity from the surface up to $z_i$ followed by a slight decrease above. Turbulence shows comparable high values only close to the surface with a clear minimum around $z_i$ where values close to the resolution limit are observed. The low variability within the profiles results partly from only two profiles with turbulence measurements in this phase. The profiles of $\mathrm{Ri}_g$ generally match the $\varepsilon$ profiles with low $\mathrm{Ri}_g$ correlating with high $\varepsilon$ but generally showing a high variability.



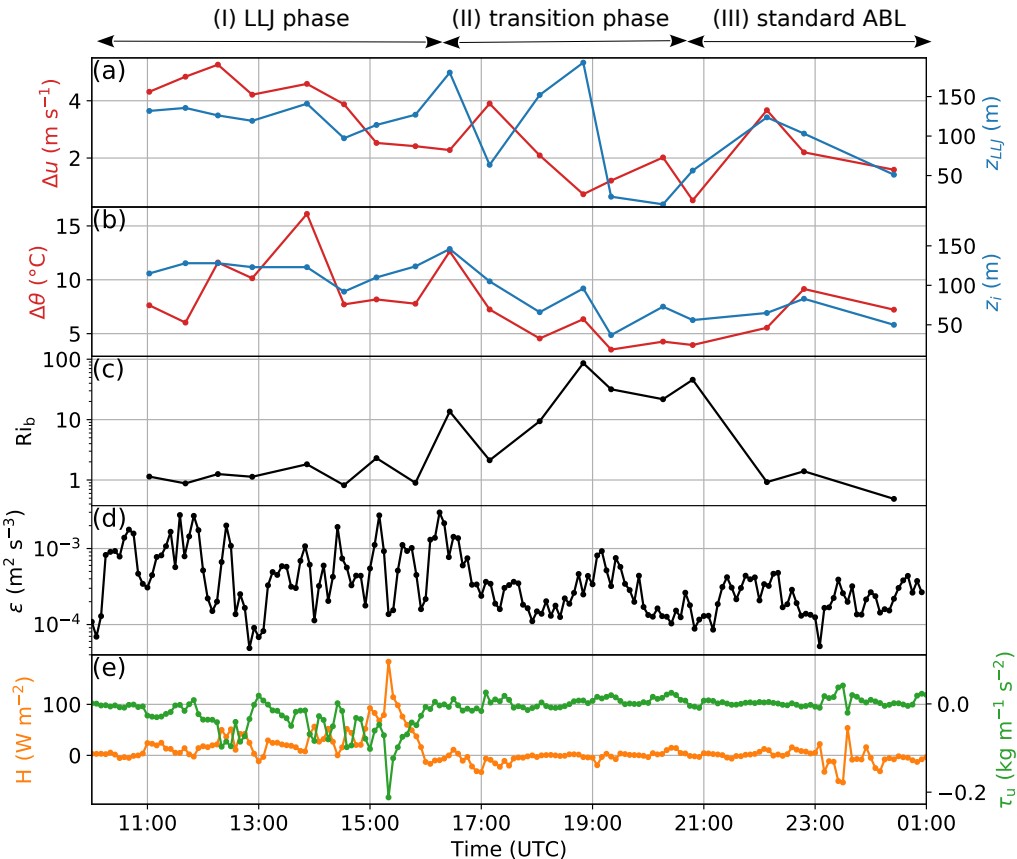

**Figure 5.** Temporal development of (a) the LLJ parameters strength $\Delta u$ and height $z_{\mathrm{LLJ}}$, (b) the temperature inversion strength $\Delta\theta$ and height $z_{\mathrm{i}}$, (c) bulk Richardson number $\mathrm{Ri}_b$ (between surface and $z_i$), (d) dissipation rate $\varepsilon$ at 65 m (5 min averages) and (e) turbulent fluxes $H$ and $\tau_{\mathrm{u}}$ at 65 m (calculated every 5 min for a 30 min period).

These normalized profiles show that the presence of an LLJ enhances turbulent mixing directly above and below the jet maximum compared to a stable ABL without an LLJ. The turbulence increase is more pronounced above the jet core. However, a stably stratified region close to the surface with height-constant wind speeds decouples the LLJ from the surface. The enhanced turbulent mixing by the LLJ might be important for the vertical mixing of advected long-range transported tracers.

## 5   The potential of an LLJ for long-range transport: an analytical modeling approach

Following the discussion in the introduction, a major motivation for this work is the question of what role an LLJ can play in the advection of a tracer - such as aerosol particles or moisture - and under which conditions this advected tracer can be mixed down to the ground despite stable stratification. Based on the observations, this question will be discussed by means of



an analytical model whose basics are briefly outlined here. A comprehensive description of the model study can be found in
Hellmuth et al. (2023).

## 5.1 Model setup

We aim to apply an analytical model for an LLJ that builds on the observations described in the previous sections. Due to
incomplete input information (i) regarding the setup of the parameters required for detailed LLJ modeling, and (ii) regarding
the meteorological and orographic drivers affecting the observations, the LLJ model employed here is reduced to one dimension
(vertical direction) with several simplifying assumptions. For this reason, the present analysis is a first estimate of the order of
210 magnitude of the expected effects on the TKE budget, and a mere qualitative assessment of the LLJ impact on local aerosol
transport.

One of the most popular theories to explain LLJs are those proposed by Blackadar (1957) describing the inertial oscillation
of an inviscid flow without explicit inclusion of frictional stress. As many studies confirmed the qualitatively correct description
of the behavior of the LLJ phenomenon by Blackadar's inertial-oscillation theory, Shapiro and Fedorovich (2010) proposed a
215 vertically continuous analytical solution of the Navier-Stokes equations within the framework of this theory. This model will
be used here in combination with an empirical adjustment of the required input parameters.

The Shapiro and Fedorovich (2010) solution describes the response of a frictional (ageostrophic) equilibrium (Ekman) flow
to a sudden reduction of eddy diffusivity, i.e. the wind transition caused by a strong impulsive reduction of the post-sunset
eddy diffusivity for a nocturnal jet. The post-sunset eddy diffusivity has been introduced into the model to emulate the sudden
reduction of the frictional stress after sunset. The model is based on the following assumptions:

- The flow is approximated by a one-dimensional homogeneous viscous incompressible pressure-driven Ekman flow over
  a horizontally homogeneous terrain.

- The ABL is assumed to be dry.

- Far above the ground ($z \to \infty$) the pressure gradient force and the Coriolis force are in geostrophic equilibrium resulting
in a spatially and temporally constant geostrophic wind.

- Near the surface, the pressure gradient force, the Coriolis force, and the frictional force are in ageostrophic equilib-
  rium. The frictional force is parameterized by a spatially constant eddy-viscosity term. The vertical eddy diffusivity is
  approximated by an "effective" value which is assumed to be constant in time and space.

- The flow is located in a right-hand Cartesian coordinate system, in which the $x$-axis is aligned with the geostrophic wind
vector $\boldsymbol{v}_g = (v_{g,x}, v_{g,y})$, i.e., $v_{g,x} = v_g = |\boldsymbol{v}_g|$ and $v_{g,y} = 0$, and the $y$-axis cuts across isobars at right angles towards
  low pressure.

- Momentum forcing originating from baroclinity is neglected.





The governing equations describing an LLJ read (Shapiro and Fedorovich, 2010):

$$\frac{\partial v_x}{\partial t} = f v_y + K_M \frac{\partial^2 v_x}{\partial z^2} \, , \tag{5}$$

$$\frac{\partial v_y}{\partial t} = -f(v_x - v_g) + K_M \frac{\partial^2 v_y}{\partial z^2} \, . \tag{6}$$

Here, $v_g = v_{g,x} \equiv |\boldsymbol{v}_g|$ denotes the geostrophic wind speed, $v_x(z,t)$ and $v_y(z,t)$ are the horizontal wind components, $K_M$ is the eddy diffusivity for momentum which relates the kinematic momentum flux to the mean velocity gradient,

$$\overline{u'w'} = -K_M \, \partial u/\partial z, \tag{7}$$

and $z$ is the height. Here, $f = 2\pi/\tau$ denotes the angular frequency and $\tau = 2\pi/f$ the oscillation period, serving as the characteristic time scale of the LLJ. Under steady-state conditions, Eq. (5) and (6) yield the known Ekman Spiral (Stull, 1997).

If the preconditions for the LLJ development are given, that is, $K_M$ is reduced by orders of magnitude due to the strong stabilization of the ABL, the nonsteady-state solution yields the Shapiro-Fedorovich helix describing the LLJ phenomenon. In the model, the reduction of the eddy diffusivity is described by $K_M = k \cdot K_{M,0}$ with $K_{M,0}$ being the eddy diffusivity of

245 momentum during the pre-LLJ phase and $k$ the turbulence reduction parameter ($k < 1$)

For extreme conditions with $K_M = 0$, Eqs. (5) and (6) can be brought into the governing equation describing a harmonic inertial oscillation of an inviscid flow:

$$\frac{\partial^2 v_x}{\partial t^2} + f^2 v_x = f^2 v_g \, , \tag{8}$$

$$\frac{\partial^2 v_y}{\partial t^2} + f^2 v_y = 0 \, . \tag{9}$$

Considering the geographical latitude (and therewith the Coriolis parameter) as given, the LLJ model has three degrees of freedom in form of the following scaling parameters:

1. the geostrophic wind speed $v_g = |\boldsymbol{v}_g|$,

2. the eddy diffusivity of momentum characterizing the pre-LLJ conditions, $K_{M,0}$, and

255 3. the turbulence reduction parameter $k = K_M/K_{M,0} < 1$.

Due to the lack of corresponding observations, the three unknown parameters are empirically adjusted to ensure a minimum relative deviation of the simulated LLJ profile from the observed LLJ. Further details of estimating those minimum-cost parameters and results are given in Hellmuth et al. (2023).



## 5.2 Simulated LLJ

Starting the LLJ simulation with the minimum-cost parameters $K_{M,0} = 3.7\,\mathrm{m^2\,s^{-1}}$ and $k = 0.098$, one obtains $K_M = k \cdot K_{M,0} \approx 0.36\,\mathrm{m^2\,s^{-1}}$. This value is well within the range of the values derived from the balloon measurements according to the following parameterization (Hanna, 1968)

$$K_M = \mathrm{C} \cdot \frac{\sigma_u^4}{\varepsilon} \tag{10}$$

with the constant $C = 0.35$.

Figure 7 displays the vertical profile of the simulated horizontal wind velocity together with the observed horizontal wind velocity averaged over the LLJ phase. While the location of the LLJ peak in the observed wind profile is well-captured by the model, the amplitude and width of the peak are not. The observed LLJ wind peak is more pronounced and narrower than the simulated one, which leads to a model underestimation of the slopes $\partial v(z)/\partial z$ (i.e., the wind shear) just below and above the altitude of the wind peak. This misprediction points toward the influence of an additional but not considered acceleration

term in the momentum budget, which may be caused by the inclination of the Greenlandic ice sheet as an orographic driver of katabatic winds. Although this physical limitation restricts the model prediction more or less to a "trend indication", the applied model is considered sufficiently substantiated to elucidate the role of LLJs in aerosol transport within the framework of a conceptual study.

## 5.3   Quantification of the contributions to the TKE budget during the LLJ phase and comparison with observations

Based on the model results, an attempt is made to estimate the order of magnitude of the process contributions to the TKE balance equation and to compare these results with observations. Assuming horizontal homogeneity, the total time rate of change of TKE is controlled by four processes (Stull, 1997):

$$\frac{\partial e}{\partial t} = \left(\frac{\partial e}{\partial t}\right)_S + \left(\frac{\partial e}{\partial t}\right)_B + \left(\frac{\partial e}{\partial t}\right)_T + \left(\frac{\partial e}{\partial t}\right)_D , \tag{11}$$

$$
\begin{aligned}
\left(\frac{\partial e}{\partial t}\right)_S &= -\overline{v_x' v_z'}\frac{\partial v_x}{\partial z} - \overline{v_y' v_z'}\frac{\partial v_y}{\partial z} , \\
\left(\frac{\partial e}{\partial t}\right)_B &= \left(\frac{g}{\overline{\theta}}\right)\overline{v_z' \theta'} , \\
\left(\frac{\partial e}{\partial t}\right)_T &= -\frac{\partial}{\partial z}\left[\overline{v_z'\left(\frac{p'}{\varrho}+e'\right)}\right] , \\
\left(\frac{\partial e}{\partial t}\right)_D &= -\varepsilon .
\end{aligned}
\tag{12}
$$

The first term on the right-hand side (subscript S) of Eq. (11) describes the TKE production due to wind shear (mechanical turbulence), the second term (subscript B) the TKE production/loss due to buoyancy (TKE generation due to positive buoyancy, TKE loss due to negative buoyancy), and the third term (subscript T) the TKE transport within the ABL, also known





as the redistribution term. This term is diffusive and vanishes when TKE is equally distributed throughout the ABL. The re-
distribution term usually comprises also a contribution to the TKE balance originating from the work of pressure fluctuations,
often associated with buoyancy or gravity waves. Finally, the fourth term (subscript D) describes the TKE loss by turbulence
dissipation into heat.

Figure 8 for the simulated LLJ phase and Fig. 9 for the averaged observations reveal that the total time rate of TKE change is
dominated by a primary loss near the surface and above the inversion layer caused by the strong dissipation at these altitudes,
and by a strong production term caused by the LLJ wind shear close to the inversion height. The buoyancy term contributes a
little to the TKE loss below the temperature inversion. The contribution of the redistribution term to $(\partial e/\partial t)$ is negligible. The
most obvious difference between the simulated and observed case is the more pronounced wind maximum inside the LLJ core
(cf also to Fig. 7) resulting in the stronger shear at this height.

### 5.4 Advection and vertical mixing

Part of the work is motivated by the question of what role an LLJ might play in the advection of a particular property such
as aerosol particles with subsequent downward mixing to the surface. Only based on the balloon measurements, this question
cannot be further treated and answered. Therefore, in the following the analytical LLJ model will be used to investigate a
possible advection of a passive tracer and under which circumstances this tracer could be effectively mixed down to the
surface.

To assess the impact of the LLJ on the transport of a passive tracer with concentration $c(x,z,t)$, here the special solutions of
the advection-diffusion equation for the LLJ situation according to the Shapiro-Fedorovich model are discussed. The following
assumptions are made:

1. The horizontal wind field is homogeneous and depends only on height $z$.

2. The airflow is assumed to be incompressible.

3. The mean vertical flow velocity vanishes ($v_z(z,t) = 0$).

4. The turbulent diffusion coefficient of the tracer equals $K_H = K_M/Pr_t = 0.14 \, \mathrm{m^2 s^{-1}}$.

5. A continuous source of the passive tracer is considered with a source strength $q_0$ in arbitrary units (a.u.) per second.

6. The $x$-axis is aligned to the geostrophic wind and advection normal to this axis is neglected.

Based on those assumptions, the advection-diffusion equation reduces to the special form:

$$\frac{\partial c(x,z,t)}{\partial t} \approx -v_x(z,t)\frac{\partial c(x,z,t)}{\partial x} + \frac{\partial}{\partial z}\left[K_H(z)\frac{\partial c(x,z,t)}{\partial z}\right] + q(x,z,t)\,,$$

$$0 \leq x \leq x_{max}\,, \quad 0 \leq z \leq z_{max}$$

(13)

with $v_x(z,t)$ given for the LLJ and pre-LLJ phase as reference.





The model is discretisized in $x$-direction (along the mean flow) according to $x_j = (j-1)\Delta x$, $j = 1, \ldots, j_{\max}$ with $\Delta x = 500\,\text{m}$ and $j_{\max} = 160$ grid points corresponding to a horizontal extension of the model domain of $x_{\max} = 80\,\text{km}$. The discretization in $z$-direction is based on $z_k = (k-1)\Delta z$, $k = 1, \ldots, k_{\max}$ with $\Delta z = 5\,\text{m}$ and $k_{\max} = 61$ grid points corresponding to a vertical extension of the model domain of $z_{\max} = 300\,\text{m}$.

Figure 10 shows the comparison of the tracer concentration $c(x, z, t)$ (in arbitrary units) between the pre-LLJ and the LLJ phase. The smaller turbulent diffusivity under LLJ conditions leads to a much weaker dilution of the passive tracer and a strong enhancement of the concentration peak in comparison with pre-LLJ conditions. In combination with the higher winds under LLJ conditions, this leads to an enhancement of the advective tracer transport into remote regions. However, the model results do not indicate that downward mixing of the tracer occurs in the LLJ phase with subsequent accumulation at the surface. This is effectively prevented by the extremely stable stratification and the resulting low turbulent diffusion constant making the advection more effective.

## 6 Summary and discussion

This study presents the observation of an LLJ based on tethered balloon measurements in the late winter central Arctic in March 2018 in northeast Greenland at the Villum Research Station/ Station Nord. The measurements span a 15-h period with a transition from a stable ABL with a prominent LLJ to a more 'normal' – but still stable – ABL without an LLJ. The observations include measurements of mean standard meteorological measurements as well as turbulence measurements from the ground to well above the ABL (typically 300 m). The balloon observations were supported by continuous turbulence measurements on a mast at 65 m height – corresponding to the lower part of the LLJ.

During the LLJ phase, observations indicate increased turbulence (increased local dissipation rates) within the LLJ, but the increased turbulence does not reach the surface. Only during the transition phase to a more classical boundary layer structure do these increased turbulence intensities also reach the surface, allowing thorough mixing to occur. These observations lead to the hypothesis that within an LLJ, a passive tracer can be 'trapped' and transported over a long distance, without vertical mixing beyond the LLJ boundaries greatly reducing the tracer concentration. Only the dissolution of the LLJ can finally lead to an increased concentration also at the ground.

To falsify this hypothesis, the analytical Shapiro-Fedorovich LLJ model was implemented and adopted to the observations. A critical point in both – the observations and the model – is that the generating drivers of the observed LLJ occurrence cannot be completely elucidated. While the occurrence of nocturnal LLJs in the mid-latitudes is caused by the onset of stabilization from the ground during sunset (by thermal radiation and consequent cooling of the ground) and associated abrupt reduction of the turbulent exchange coefficient, in our study an influence of the Greenland ice sheet and associated catabatic cold air outflows cannot be excluded as a cause. However, the exact cause for the reduction of the turbulent exchange coefficient and the subsequent occurrence of the LLJ is not of fundamental importance to our discussion. The model is able to qualitatively reproduce the vertical profile of the horizontal wind during the LLJ. Based on these modeled vertical profiles, the advection of a passive tracer is qualitatively simulated for the case with and without LLJ. The results support the idea of a possible advection



of a passive tracer over larger distances with subsequent downward mixing during the collapse of the LLJ. However, within the framework of the present study we cannot answer over which distances such an advection can extend.

One can hypothesize that the same is true for the advective transport of possible reactive tracers that may be involved in the formation of precursor gases for aerosol particle formation such as found by Siebert et al. (2007). Consequently, LLJs may have a significant impact on both in-situ and ex-situ aerosol formation and evolution under Arctic conditions, which requires

further investigation through appropriate observations of the parameters involved, including vertical profiling.

*Data availability.* The observational data for this manuscript are publicly available (Egerer et al., 2019).

*Author contributions.* UE and HS performed the balloon measurements and processed and analyzed the data. LLS contributed the meteorological mast measurements. OH set up, ran and analyzed the analytical model. All co-authors contributed to drafting the manuscript.

*Competing interests.* The authors declare that they have no conflict of interest.

*Acknowledgements.* We gratefully acknowledge the funding by the Deutsche Forschungsgemeinschaft (DFG, German Research Foundation) - project number 268020496 - TRR 172, within the Transregional Collaborative Research Center "ArctiC Amplification: Climate Relevant Atmospheric and SurfaCe Processes, and Feedback Mechanisms (AC)[3]" in sub-project A02. We acknowledge the essential help of Andreas Herber, Jens Voigtländer, Frank Stratmann, PAMARCMiP scientists and the crew of Station Nord and Villum Research Station during the balloon operations. Sven-Erik Gryning provided and advised us on ceilometer and wind lidar data at VRS. We wish to thank Andreas

Massling, Daniel Charles Thomas, and Jakob Boyd Pernov for discussing VRS data.



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







**Figure 6.** Normalized vertical profiles of $u$, $\theta$, $\varepsilon$, $\mathrm{Ri}_g$ and $q$ for phase I (with LLJ, top), phase II (transition, center) and phase III (profiles without LLJ, bottom). The box plots show variations within the individual $N$ profiles in each phase. The height $z$ is normalized with the temperature inversion base height $z_i$. The wind velocity is normalized with $u_{\max}$ ($u_{\mathrm{LLJ}}$ or maximum $u$ below 300 m). The quantities $\theta$ and $q$ are normalized with their value at $z_i$. The boxes include the lower and upper quartile values of the data, with the orange line at the median.



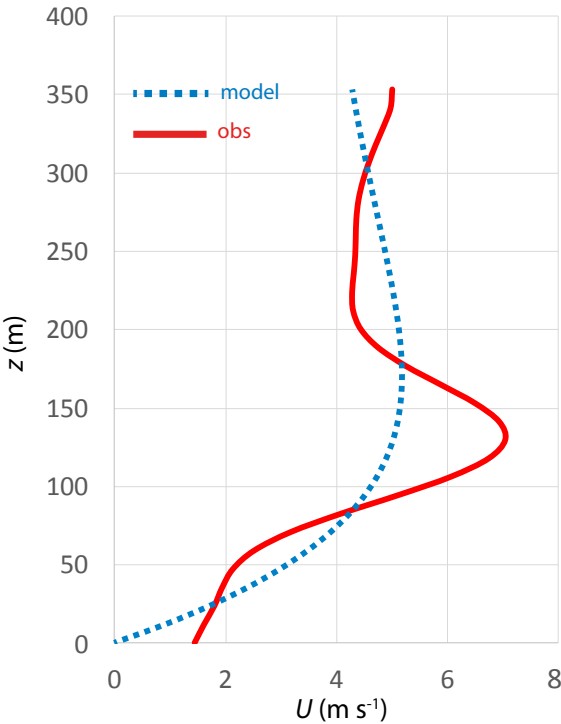

**Figure 7.** Vertical profile of the simulated horizontal wind velocity (blue dotted curve) together with the observed horizontal wind velocity during the LLJ phase (solid red curve).





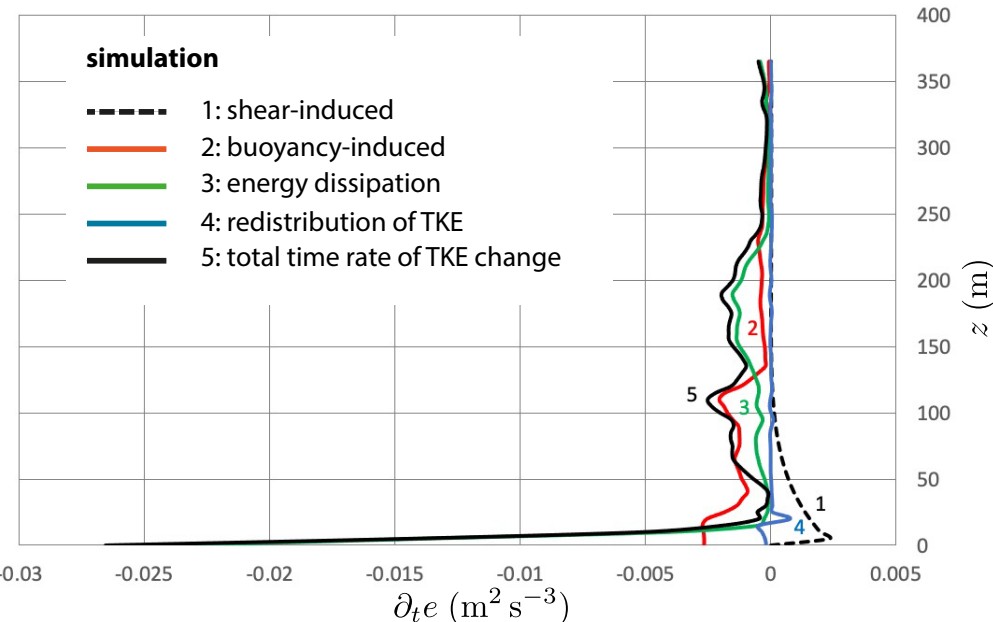

**Figure 8.** Vertical profiles of the simulated contributions to the TKE budgets: shear-induced contribution $(\partial e/\partial t)_S$ (graph 1, dotted black), buoyancy-induced contribution $(\partial e/\partial t)_B$ (graph 2, red), dissipation-induced contribution $(\partial e/\partial t)_D$ (graph 3, green), redistribution contribution $(\partial e/\partial t)_T$ (graph 4, light-blue), total time rate of TKE change $(\partial e/\partial t)$ (graph 5, black).



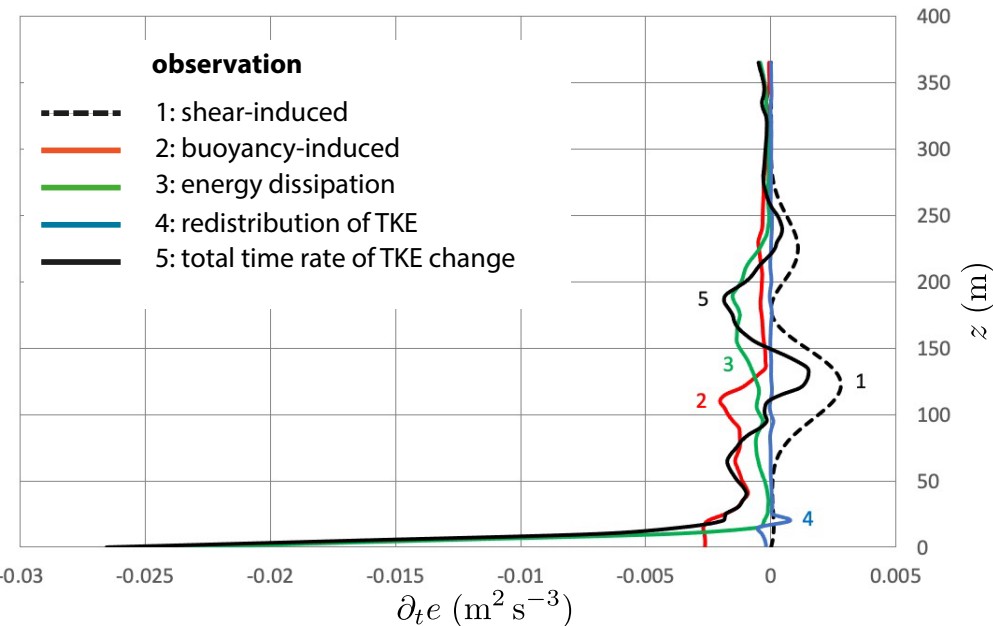

**Figure 9.** Vertical profiles of the observed contributions to the TKE budgets: shear-induced contribution $(\partial e/\partial t)_S$ (graph 1, dotted black), buoyancy-induced contribution $(\partial e/\partial t)_B$ (graph 2, red), dissipation-induced contribution $(\partial e/\partial t)_D$ (graph 3, green), redistribution contribution $(\partial e/\partial t)_T$ (graph 4, light-blue), total time rate of TKE change $(\partial e/\partial t)$ (graph 5, black).



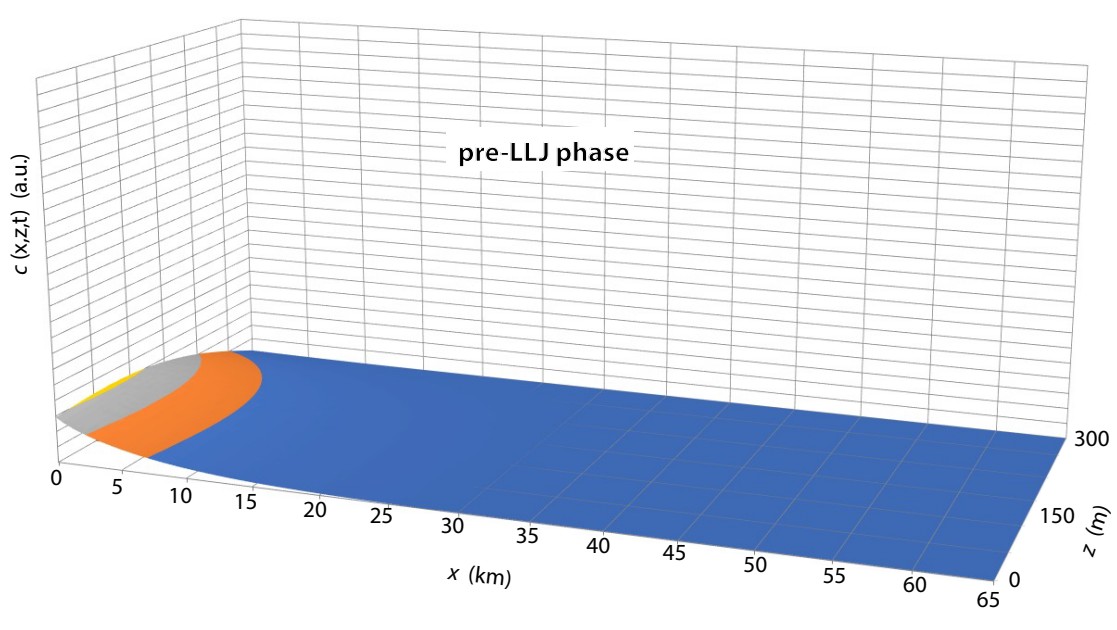

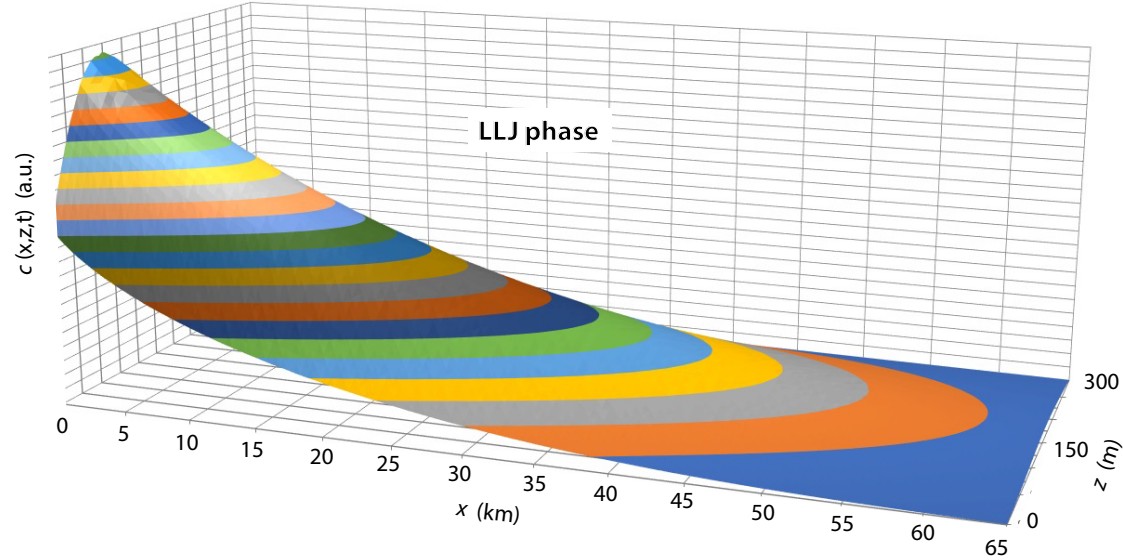

**Figure 10.** Tracer concentration $c(x, z, t)$ in arbitrary units (a.u.) as function of $x$ up to 65 km (along the mean flow) and the vertical $z$ up to 300 m at dimensionless time $T$=3. Upper panel: pre-LLJ phase (Ekman helix); lower panel: LLJ phase (Shapiro-Fedorovich model).