# Peer review of "The role of a low-level jet for stirring the stable atmospheric surface layer in the Arctic"

_EGUsphere, 2023_

## Referee Comment (RC2)

**The role of a low-level jet for stirring the stable atmospheric surface layer in the Arctic**

Ulrike Egerer, Holger Siebert, Olaf Hellmuth, and Lise Lotte Sørensen

This paper presents novel observations of the atmospheric surface layer over northeast Greenland in March 2018, using ground-based and balloon-borne probes. The vertical profiles over a 15 hr period show the occurrence and disappearance of a low level jet. The observations are generally clearly presented and are of interest to ACP readers. The authors go on to propose a hypothesis, that the LLJ can act to advect pollutants above the surface, which can then be mixed back to the surface when the LLJ ceases. The data presented make this a reasonable hypothesis, but the paper tries to use a model calculation to examine the hypothesis further. I did not find this part of the paper convincing: the model is very crude and I could not understand the presentation of the results (fig 10). I can see the value of trying to compare the observations to the 1-D model, but given the disparities I do not see the value of using the model to examine the hypothesis. For sure, if you reduce the diffusion coefficient such that tracer released into a stable layer cannot reach the surface, advection will move the tracer downstream. But I don't see how this adds value to what can be deduced from the observations themselves, because the LLJ conditions are manifestly not 1-D.

Detailed comments:

l. 28. Richardson number (not Richards)

l.90, eqn 2. Suggest you write $(\Delta u(z))^2$ since $\Delta u^2(z)$ could be confused with taking the difference in $u^2$

l.94. Please provide appropriate references for equation 3. Siebert et al (2006) makes no mention of structure functions, and Wyngaard's book does not present the t*u formulation used in this paper. Also, it is not clear to me why you use the same averaging period (2s) for the structure function and for the mean velocity.

l.101, equation 4. Again, please give a suitable reference – I cannot find it in Wyngaard's book but if it's there please provide the page number. There is on p. 16 the expression $\varepsilon \sim u^3/\ell$ referring to the largest (energy-containing) eddies, so I presume this is what is meant (this is also implied by l. 110). Turbulence is a phenomenon with a cascade of motions from the energy-producing scales to those of viscous dissipation. One could even say that a defining characteristic of turbulence it that it doesn't have a typical scale!

l.102. You say that $\varepsilon$ is derived over 5 s scales but on l.96 you say 2 s. In any case, why are you performing a regression of two quantities evaluated over different scales ($\sigma$ is smoothed over 30 seconds)?

L.105, 108. figure caption says $Ri_g$, not $Ri_b$. Which is it?

l.106. The only length scale that can be sensibly deduced from fig 2 is 4 m, from the regression line. I presume that by 'local length scale' you mean the ratio of individual $\sigma^3/\varepsilon$ values, but this will be severely affected by stochastic noise.

l.112. analogously

fig.3 caption: ascents and descents

l.185. Looking at fig. 6 I can only see two maxima, one at the surface and the other at $z/z_i \sim 1.5$, not three as the paper claims. Likewise, there isn't a minimum at $z=z_i$.

l.288. The paragraph as written implies that the model (fig 8) and the observations (fig 9) agree, but they clearly don't. The observations show peaks in the TKE production at 130 and 240 m arising from the shear-induced peaks at 125 and 225 m (the inversion height is, I presume from fig.5, at 100 m), but the model shows a peak near the ground. The red, green and blue curves in the two plots are identical so have clearly been plotted twice – from which source I can't tell. The correct versions of these curves need to be plotted and the paragraph re-written to describe what is actually shown on figs 8 and 9.

Section 5.4. It would help here if the profile of $K_m$ from the observations (derivable from equation 10) were shown and compared with the assumed value in the model. This will show whether the diffusion calculation in the model is at all useful as an estimate of diffusion in the real case.

l.306. As far as I can make out, $K_m$ in the LLJ simulation should be $kK_{m,0} = 0.098*3.7 = 0.362 \ m^2s^{-1}$ (l.260). Then a value of Pr=2.59 must be assumed to get the quoted $K_H$. Why did you choose this value? The paper should justify this choice. What is the equivalent $K_H$ in the pre-LLJ atmosphere? With the lower stability in this case Pr should be around 1 so would $K_H \sim K_{M,0}$?

L.307 At what height is tracer released in the model? Is there a continuous source at x=0 and at some height (or height range)? Without this information it is not possible to understand fig.10.

Fig 10. A much more informative caption is needed for this bewildering figure. What is the colour scale? Is the dark blue that dominates the top panel a low or a high concentration? Do the two plots show maxima in concentration descending towards the ground? Why are there only four colours on the upper plot and 23 on the lower plot? Where is the source of tracer? Is it the same in the two cases?

L.340 katabatic

---

## Author Comment (AC1)

**Author comments to reviewer 1**

We appreciate anonymous reviewer #1's feedback on our manuscript and are grateful for the time and effort for the review. Addressing the helpful review comments remarkably improved the quality of the manuscript. We respond to the individual comments below in blue color.

**General Comments**

Overall, the paper is very well-written and the analysis is very relevant for improving our understanding of the relationship between LLJs and the ABL in the Arctic, which is crucial for the improvement of global climate models and the enhancement of knowledge regarding the interaction between the surface and overlying atmosphere in the central Arctic.
Thank you, we really appreciate this feedback.

To improve this paper, I suggest the authors add some more description of a few aspects of the methods, some chosen terminology, and some figures, as discussed in the minor comments below.
We address the individual comments on the observational part as discussed below. However, leveraging the feedback from the other reviewer, we decided to remove the model part (former Sect. 5) from the paper entirely. Taking into account the critical points, we see that the simple model study does not add sufficient value to the observations. For a detailed explanation, please see our response to the other reviewer's comments.

**Minor Comments**

L13: It should be noted that often the LLJ is above the ABL, rather than always being within it. This can be particularly evident when the LLJ is decoupled from the surface in the case of a stable LLJ, and the formation mechanism is inertial oscillations.
We agree with this clarification and modified the sentence accordingly: "An LLJ , especially in conjunction with stably stratified boundary layers, is usually found in the upper region of the ABL or even partly just inside or above the inversion."

L57: Please include some uncertainties for the two sensor packages described in (i) and (ii).
We agree that a statement about accuracy and resolution of our measurements is missing in the manuscript and include the following sentences into the manuscript: "The absolute accuracy of the wind speed measurement with the Pitot-static tube is determined by the accuracy of the differential pressure sensor and is additionally inversely proportional to the wind speed itself. For typical wind speeds of 5 m/s, the absolute accuracy is only in the range of 0.5 m/s which is essentially noticeable by a zero offset and can therefore be partially corrected; however, the relative resolution is higher by a factor of 10 for the Pitot-static tube and the spectrally determined resolution of the hot-wire anemometer is in the range of 0.2 cm/s (Egerer et al., 2019)."

L60: It is mentioned that the ABL transitioned to being a "more classic stable ABL." You should include a characterization of the ABL before the transition, so it is well-understood how the ABL changed (aside from just the presence to dissipation of the LLJ).
We agree with the suggestion to describe in more detail what we understand by a classical boundary layer structure and therefore extend the sentence as follows: " ... more classically stable ABL which is characterized by the wind speed increasing mostly logarithmically from the ground with height. The turbulence is generated by the wind shear and is therefore strongest near the ground and decreases continuously with height, depending on the damping effect of the temperature inversion."

Figure 1: I am a little confused by panel (a). Please specify what StdMeteo and Hotwire refer to – are these the two sensor packages described in L52-57? You should clarify the abbreviations in the figure legend. Also, it appears the BELUGA doesn't always fly with both sensor at the same time? Please also clarify why this is and how it was determined in the field that certain profiles only had one or the other sensor in this case study.

Yes, the abbreviations refer to the instrument packages, we added that information in the Figure caption. Also, we added the following explanation in the text: "The instrument packages were deployed in a rotating sequence aboard BELUGA to ensure adherence to the payload limit; a flight overview table with deployed instruments is provided in Egerer et al. (2019b)."

L75-78: Your criteria for defining LLJs differ some from the standard literature (e.g., the requirement of the LLJ core being below 250 m, and the LLJ strength being relative to the higher value of the wind minimum above and below the core). Please add some explanation about why you divert from the more standard criteria for an LLJ. For example, include mention of any testing you might have done to determine that these methods provide valid results.

A widely used LLJ definition is the one by Andreas et al. (2000) as follows: "If the wind speed profile shows a local maximum that is 2 m/s higher than speeds both above and below it, we call the feature a jet. Notice, with this definition, the jet must be elevated and cannot occur at the surface. This definition is similar to Stull's (1988, p. 521), except he does not require the jets to be elevated." We modified the Andreas et al. (2000) criteria to exclude profiles that objectively do not resemble an LLJ; two examples are shown in Fig. 1 of this document. Our criteria are tailored to our specific case and we don't argue that they should be used universally. Reasons for modifications are:

- wind speed difference above *and* below as in Andreas et al. (2000)

- wind speed difference of 2.5 m/s instead of 2 m/s because our profiles are at a high resolution and small-scale fluctuations contribute to the wind speed difference (right example in Fig. 1).

- height below 250 m because we want to continuously study a specific, consistent LLJ at the lowest level.

We add in the text: "We tailor some of the criteria used in the literature for defining LLJs to our specific observations and the unique characteristics of this particular LLJ."

[Figure]

Figure 1: Example LLJ profiles that we objectively do not consider as LLJs.

L102: Add a space between 5 and s in "5s-long"
Done.

Figure 3: Rather than "the ascends and descends are from the BELUGA" perhaps say "The time-height profiles are from the BELUGA."
Done.
Also, for consistency, it would be better to include all temperature measurements in the same units, either temperature or potential temperature, rather than a mix of both.
We argue that it makes sense to display the vertical profiles and constant-height time series in different temperature representations. For vertical profiles, the potential temperature provides the benefit that thermodynamic stratification is directly visible based on the gradient. In contrast, it makes no sense to transform temperature at a constant height to potential temperature, even if the difference is negligible at the low altitudes shown. We therefore kindly ask the reviewer to accept our explanation and leave the temperature presentation as it is.

L122: Somewhere (maybe in the Intro?) you should define what you mean when you say "standard stable ABL" for northern Greenland, as this could be different depending on location, and it shouldn't be assumed that a reader would know what you mean. Perhaps use wording such as "<description of standard ABL>, which will hereafter be referred to as a "standard stable ABL."
Agreed that we did not define what we mean by standard stable ABL. We added: "By a "standard stable ABL" we refer to a cloud-free, shallow stable ABL in which terrestrial radiation causes a surface-based temperature inversion, comparable to a nocturnal boundary layer at mid-latitudes."

L130: By "strong surface temperature gradient" do you mean strong surface temperature inversion? Please specify.
Yes, we mean strong surface temperature inversion. We changed this in the text.

Referee's comments to L297, L312, L314, and Fig. 10
These comments refer to Section 5 of the originally submitted manuscript, which addresses the LLJ modeling approach. Referee 2 did not find the modeling part of the paper convincing because the model is very crude, does not add value to what can be deduced from the observations themselves, and is not able to describe the LLJ conditions which are manifestly not 1-D. As the data availability does not allow the setup of a 3D model study we decided to remove this section from the revised manuscript. Nonetheless, the comments of referee 1 on this part were found absolutely correct and helpful. We are grateful!

L326: In some central Arctic locations, an LLJ is almost ubiquitous, so it is rather the more "normal" case that a stable ABL co-occurs with an LLJ. Please add some discussion (probably in the Intro) of literature which leads to the conclusion that in northeast Greenland, it is more common to not have an LLJ with a stable ABL.
We agree that for northeast Greenland, it is hard or even impossible to define a "normal" ABL. By "normal", we mean the classical textbook nocturnal ABL as mentioned in comment l. 122. We change the wording in l. 326 to "classically stable nocturnal ABL". Additionally, we add some more LLJ characterization for the study area in the introduction.

---

## Author Comment (AC2)

**Author comments to reviewer 2**

We thank Geraint Vaughan for the time and effort to review our manuscript and for the constructive comments. Following the reviewer's suggestion, we carefully evaluated the former Sect. 5, the analytical modeling approach, and concluded that the model does not provide sufficient value to support the observations, as discussed in detail below. Our responses to individual comments are outlined below, highlighted in blue color.

**General**

[...] the paper tries to use a model calculation to examine the hypothesis further. I did not find this part of the paper convincing: the model is very crude and I could not understand the presentation of the results (fig 10). I can see the value of trying to compare the observations to the 1-D model, but given the disparities I do not see the value of using the model to examine the hypothesis. For sure, if you reduce the diffusion coefficient such that tracer released into a stable layer cannot reach the surface, advection will move the tracer downstream. But I don't see how this adds value to what can be deduced from the observations themselves, because the LLJ conditions are manifestly not 1-D.

We thank the referee very much for his careful evaluation and clear statement. Owing to the lack of data required for the specification of the initial and boundary conditions, a 3D modeling study could, unfortunately, not be conducted within the framework of the present phenomenological study. Despite the known restrictions associated with the application of a 1D modeling approach to high-Arctic LLJs we decided to add to the present study at least a separate analysis, in which available observational data were merged with qualified empirical a-priori information on ABL turbulence to simulate an LLJ within the framework of Blackadar's inertial-oscillation theory. The underlying assumptions, parameterizations, and steps of data evaluation are described together with first-guess estimates of the TKE-budget terms in a very detailed manner in a separate non-peer-reviewed document (Hellmuth et al., 2023). Our main intention, however, was to set up a conceptual model to demonstrate (and quantify) the impact of an LLJ on the long-range transport of a passive tracer. This was clearly achieved. In any case, it speaks for the referee's meteorological expertise and experience when he does not see "a need" for a model-based proof of LLJ potential for long-range tracer transport. It is very fine with us if this aspect is considered sufficiently substantiated by the presented observations. Thus, in response to the referee, we decided to remove former Section 5 from the paper. At the same time, we hope that the referee accepts the short discussion of the model in the "summary and discussion" section, referring to the above-mentioned non-peer-reviewed study.

**Detailed comments:**

l. 28. Richardson number (not Richards)
Changed.

l.90, eqn 2. Suggest you write $(\Delta u(z))^2$ since $\Delta u^2(z)$ could be confused with taking the difference in $u^2$
Changed.

l.94. Please provide appropriate references for equation 3. Siebert et al (2006) makes no mention of structure functions, and Wyngaard's book does not present the t*u formulation used in this paper.
There might be a misunderstanding but the structure-function approach for estimating $\varepsilon$ in Siebert

et al. (2006) is explained in detail with equations 10 and 11 therein. These equations also make use of the Taylor transformation (t*u). We add the equation numbers and remove the Wyngaard reference in this context.

Also, it is not clear to me why you use the same averaging period (2s) for the structure function and for the mean velocity.

The dissipation as estimated from the structure function is interpreted as an instantaneous value "valid" for the 2s period and, therefore, the Taylor transformation should be performed over the same integration period as described around Eq. 10/11 in Siebert et al. (2006).

l.101, equation 4. Again, please give a suitable reference – I cannot find it in Wyngaard's book but if it's there please provide the page number. There is on p. 16 the expression $\varepsilon \sim u^3/\ell$ referring to the largest (energy-containing) eddies, so I presume this is what is meant (this is also implied by l. 110). Turbulence is a phenomenon with a cascade of motions from the energy-producing scales to those of viscous dissipation. One could even say that a defining characteristic of turbulence it that it doesn't have a typical scale!

Yes, we refer to the largest (energy-containing) eddies, this is changed in the text now and we added the page number. Although Wyngaard's original definition refers to a mean value for $\ell$, the book also mentions epsilon intermittency, and this is what we refer to by our locally defined $\ell$.

l.102. You say that $\varepsilon$ is derived over 5 s scales but on l.96 you say 2 s. In any case, why are you performing a regression of two quantities evaluated over different scales ($\sigma$ is smoothed over 30 seconds)?

We agree that using 5 s instead of 2 s for $\varepsilon$ (as used otherwise in this paper) was confusing. Earlier studies (e.g. Egerer et al., 2019 and Siebert et al., 2006) showed that the estimation of dissipation rates is insensitive to the choice of the averaging time, but smaller time windows provide a better time resolution and can reflect intermittency. However, as suggested by the reviewer, for this plot we now average $\varepsilon$ over 30 s to be consistent with the $\sigma$ calculation. Now, each dot in Fig.2 represents a 30 s time interval. This adjustment does not change our message about derived length scales and the fit parameters of the regression do not change significantly.

L.105, 108. figure caption says $Ri_g$, not $Ri_b$. Which is it?

Please excuse the typo, the figure correctly shows $Ri_g$, we changed this in the text.

l.106. The only length scale that can be sensibly deduced from fig 2 is 4 m, from the regression line. I presume that by 'local length scale' you mean the ratio of individual $\frac{\sigma^3}{\varepsilon}$ values, but this will be severely affected by stochastic noise.

By "local length scale" we mean the ratio of individual, locally-fluctuating $\sigma^3/\varepsilon$ values, we added this explanation in the text (see also comment l. 101.). We interpret the "scatter" in Fig. 2 not as stochastic noise, but instead as naturally occurring, short-lived fluctuations of observed turbulent length scales. Of course, we are aware that this is not so simple, because although the dissipation rate can be interpreted as a local lognormally distributed quantity according to K62, this interpretation is not directly applicable to $\sigma$. However, we think that this discussion may go a little bit beyond the main topic of this manuscript. With Fig. 2, we aim to show the range of these length scales and how they relate to the different turbulent parameters $\varepsilon$, $\sigma$ and $Ri_g$.

l.112. analogously

Changed.

fig.3 caption: ascents and descents

Changed.

l.185. Looking at fig. 6 I can only see two maxima, one at the surface and the other at z/zi    1.5, not three as the paper claims. Likewise, there isn't a minimum at z=zi.

We agree that some patterns in the profile were over-interpreted and changed the text to: "In the LLJ period, the average $\varepsilon$ structure has a characteristic shape with two local maxima: near the surface and at $z/z_i \approx 1.5$ with reduced values around $z_i$ itself. A local minimum of $\varepsilon$ at $z/z_i \approx 0.3$ suggests decoupling of the LLJ from the surface."

All subsequent comments referring to former Sect. 5 - the analytical modeling approach:

Referee's comments on Section 5, especially on L288 (figs. 8 & 9), L306-307, and on fig. 10 are absolutely correct and helpful. A correct explanation of what has been presented in the incriminated figures can be found in the non-peer review study cited in the summary and discussion section. We do not respond to the comments here, since this section is removed in the new version of the manuscript.